# OpenReview forum: "Position: LLMs Should Incorporate Explicit Mechanisms for Human Empathy"
_ICML.cc/2026/Position_Paper_Track — ICML 2026 Position Paper Track regular_

### Official Review · Reviewer_xUJY · 2026-02-14

**Significance:** 3
**Argument Clarity:** 3
**Rating:** 5
**Confidence:** 4

**Questions:**

see weakness

**Alternative Views Section:**

Yes

**Compliance With Llm Reviewing Policy A Conservative:**

Affirmed.

**Discussion Potential:**

3

**Paper Summary:**

The paper proposes and systematically summarizes four common empathy failure mechanisms in current LLMs: emotional attenuation (flattening expressions of urgency, pain, moral weight, etc.), empathy granularity mismatch (too coarse or too detailed in addressing sensitive content), conflict avoidance (replacing genuine tension clarification and discussion with premature advice or pursuit of superficial harmony), and linguistic detachment (using abstract, depersonalized, or passive expressions to create interpersonal distance). Corresponding mechanistic metrics are also provided to directly capture these distortions in evaluation. The author also introduces the "Three-Dimensional Empathy" framework to explain the varying performance of empathy demands across different tasks: cognitive empathy focuses on retaining key information and inferences of meaning, cultural empathy emphasizes the retention of cultural symbols, style, and aesthetic context, and relational empathy prioritizes maintaining trust, cooperation, and psychological safety in interactions.

To support this stance, the author conducts an empirical analysis of ten state-of-the-art models across three high-risk tasks: news generation (cognitive empathy), literary translation (cultural empathy), and psychological counseling responses (relational empathy). The results show that while the models often perform well on traditional metrics such as ROUGE, COMET, semantic similarity, or counseling style scores, they still exhibit stable and systematic distortions in empathy mechanism indicators. For example, the sense of urgency in news text is weakened, the narrative voice and cultural tone in literary translation are "homogenized," and there is widespread emotional flattening and conflict avoidance in counseling dialogues. This suggests that existing benchmarks may obscure alignment trade-offs directly related to human experience.

**Position:**

Yes

**Position In Title:**

Yes

**Related Work:**

3

**Strengths And Weaknesses:**

The paper breaks down empathy failures into four reusable mechanisms (emotional attenuation, granularity mismatch, conflict avoidance, linguistic detachment) and attempts to provide quantifiable metrics (SA/EGM/CA/LD). This decomposition, compared to simply saying "the model lacks empathy," is more like a "locatable bug report," which is inspiring for future research (data, alignment objectives, and evaluation set design).

The distinction between cognitive, cultural, and relational empathy, along with the task-mapping table, explains why the same "lack of empathy" manifests differently across tasks such as news generation, literature translation, and counseling. This framework also encourages community discussions: which tasks rely more on which type of empathy and how should we evaluate and balance them.

The paper reports both standard and mechanism-specific metrics across three task types, demonstrating the phenomenon where "semantic similarity/translation metrics/counseling style scores are high, but empathy mechanism metrics still show significant distortions." This supports the argument that existing evaluations may not capture the fidelity of human perspectives.


## Weakness

SA relies on an "emotional intensity scorer" (E(\cdot)), EGM depends on an "inappropriate content counter" (U(\cdot)), CA relies on a "conflict avoidance discriminator" (J(\cdot)), and LD relies on "linguistic detachment feature extraction." If these components' biases, cross-domain stability, and consistency with human judgments are not sufficiently validated, there may be issues such as:

* The metric capturing a "writing style change" rather than a true empathy distortion.
* Scorers in different cultures/genres (especially in literary translation) may have systematic biases.
* CA, as a binary classifier, especially depends on how the discriminator is defined, and the boundary examples may be highly controversial.

Currently, the evidence suggests "the metrics show issues," but it is not enough to prove that "humans would indeed perceive this as empathy distortion" or that "distortion leads to worse human outcomes."

Small sample sizes and selection bias: For the news task, 50 samples were randomly selected after filtering high-impact samples by LLM; the counseling task also used a dataset of 50 samples. With a small sample size and an opaque filtering process, there is a risk of selection bias, which may affect the robustness and reproducibility of the conclusions. However, given that this is a position paper, this is acceptable.

Misalignment with real-world applications in the task setup:

* News generation: Given the "gold standard summary" to generate a full news article, this setup resembles a controlled generation experiment and may not reflect real-world news generation or rewriting processes.
* Counseling: Lacks reference therapist responses, relying solely on EPITOME and custom mechanism metrics, making it difficult to verify the role of "relational empathy" in real multi-turn counseling.
* Literary translation: Using the WMT literary translation task is reasonable, but the boundary between "cultural empathy" and general stylistic fidelity or translation naturalness needs to be more clearly defined.

Limited discussion on practical optimization strategies: Although the paper proposes "empathy as an alignment objective, benchmark design, and data planning" in three directions, it provides little discussion on concrete feasible optimization strategies (e.g., how to weight in multi-objective RLHF, how to avoid conflicts with safety objectives, and how to construct controllable training signals). While this is acceptable for a position paper, it falls short if the goal is to drive practical implementation.

**Support:**

3

---

> ### Author Rebuttal · Authors · 2026-03-31
>
> Thank you for recognizing our measurable decomposition of empathy failures, the task-based empathy framework, and the limits of standard metrics.
>
> > W1: Insufficient validation of metric components and alignment with human judgment.
>
> Thank you for this important concern.
> We agree that validating bias, cross-domain robustness, and alignment with human judgments is crucial.
> As a position paper, our goal is to introduce a **mechanism-grounded framework** and provide initial evidence that these distortions are measurable, rather than to claim fully mature or exhaustive validation.
>
> Our metrics are not based on free-form judgments: each uses **explicit scoring rubrics**, with judge LLMs constrained to predefined criteria.
> To assess robustness, we repeat evaluations with Claude-Sonnet-4.5 and Gemini-3-Pro (**Table T1, see JCeE**), observing consistently high inter-judge agreement (Spearman: SA 0.835-0.948, CA 0.748-0.924).
> While absolute scales vary, relative conclusions remain stable.
>
> We will clarify this scope in the revision and position these metrics as *diagnostic probes* rather than definitive measures. We also agree that large-scale human validation and cross-domain calibration are essential next steps and will highlight them as future work.
>
> > W2: Small sample sizes and selection bias.
>
> Thank you for this observation. We agree that a small sample size and filtering may introduce bias and affect robustness. The current 50-sample setting is indeed limited.
>
> To assess this, we expand the counseling dataset to 100, 150, and 200 examples. As shown in **Table T3**, results become significantly more **stable** with larger samples, exhibiting low variance and Spearman correlations across sizes approaching 1, indicating consistent model rankings. While some variance remains (e.g., for Claude-Sonnet-4.5 in SA), this highlights the limitation of small samples. We will include this sensitivity analysis and clarify the filtering process in the revision to improve transparency and reproducibility.
>
> **Table T3**: *Psychological counseling results for relational empathy under different dataset sizes on the Sentiment Attenuation (SA) metric.*
>
> |Model|50|100|150|200|
> |-|-|-|-|-|
> |GPT-5.2|0.529 ± 0.011|0.478 ± 0.009|0.491 ± 0.007|0.488 ± 0.007|
> |Claude-Sonnet-4.5|0.481 ± 0.012|0.532 ± 0.007|0.535 ± 0.007|0.542 ± 0.008|
> |Gemini-3|0.512 ± 0.003|0.366 ± 0.042|0.363 ± 0.050|0.368 ± 0.046|
> |Grok-4.1|0.515 ± 0.015|0.423 ± 0.015|0.439 ± 0.013|0.431 ± 0.014|
> |Llama-4|0.529 ± 0.005|0.514 ± 0.002|0.513 ± 0.002|0.511 ± 0.002|
> |DeepSeek-V3.2|0.530 ± 0.008|0.530 ± 0.015|0.497 ± 0.009|0.495 ± 0.010|
> |Qwen3-VL|0.431 ± 0.046|0.303 ± 0.055|0.293 ± 0.054|0.294 ± 0.054|
> |Kimi-K2|0.487 ± 0.063|0.344 ± 0.069|0.356 ± 0.064|0.365 ± 0.057|
> |GLM-4.7|0.509 ± 0.050|0.416 ± 0.041|0.425 ± 0.035|0.425 ± 0.032|
> |Doubao-Seed1.8|0.337 ± 0.028|0.212 ± 0.043|0.208 ± 0.043|0.210 ± 0.045|
>
> > W3: Misalignment with real-world application in task setup.
>
> Thank you for this thoughtful comment. We agree that these tasks are simplified setups rather than full real-world deployments.
> Our goal is to use **controlled settings** to isolate and measure each empathy dimension, with broader applications discussed in Table 1.
>
> For **news generation**, using gold summaries provides a controlled way to test whether models preserve urgency and human impact.
> For **counseling**, the setup serves as an initial probe of relational failures, not a full multi-turn simulation.
> For **literary translation**, we will clarify the distinction from general stylistic fidelity, emphasizing preservation of culturally meaningful tone and context.
> We will make these scopes and limitations more explicit in the revision.
>
> > W4: Limited discussion of practical optimization strategies.
>
> Thank you for this helpful point. We agree that concrete optimization strategies are important for practical deployment. As a position paper, our goal is to establish empathy as a **measurable alignment objective** and outline directions in objectives, benchmarks, and data.
>
> We view detailed optimization (e.g., multi-objective weighting, safety trade-offs, controllable signals) as a natural next step built on this framing. We will clarify this and briefly outline potential directions to better connect our proposal to practical implementation.

---

> > ### Author Rebuttal · Reviewer_xUJY · 2026-04-03
> >
> > I select option (a). The authors have adequately addressed all my key concerns in their rebuttal.

---

> > > ### Author Response · Authors · 2026-04-03
> > >
> > > We sincerely thank the reviewer for the positive feedback and for confirming that the concerns have been fully addressed.
> > >
> > > We greatly appreciate your thoughtful comments throughout the review process, which have helped us improve the clarity, positioning, and overall quality of the paper.

---

### Official Review · Reviewer_1X8y · 2026-03-11

**Significance:** 3
**Argument Clarity:** 2
**Rating:** 3
**Confidence:** 5

**Questions:**

1. Please justify the definitions of the four metrics. For example, why GM uses (a - b) / (a + b) instead of (a - b) / a, similar to SA?

**Alternative Views Section:**

Yes

**Compliance With Llm Reviewing Policy A Conservative:**

Affirmed.

**Discussion Potential:**

3

**Paper Summary:**

This paper advocates to explicitly introduce human empathy mechanisms into LLMs, instead of treating empathy as a side product of AI alignment. The authors define empathy as a measurable and observable behavior: models can model human's views, intentions, emotions, and demands when generating outputs. The paper includes a detailed literature review of empathy-critical LLM applications. The paper also provides preliminary analyses that models without empathy can generate undesired outputs.

**Position:**

Yes

**Position In Title:**

Yes

**Related Work:**

4

**Strengths And Weaknesses:**

Strengths
- The paper falls into interests by researchers from LLM alignment, post-training, LLM application, evaluation, and HCI.
- Timeliness: more users are treating AI as friends beyond mere tools.
- The stated position brings benefits of improving model ability to do personalized ToM and context-sensitivity.

Weaknesses
- In the news domain, overemphasizing emotional resonance may lead to the sensationalization of objective facts or even the introduction of subjective bias.
- Anthropomorphism may mislead users into believing that machines possess real emotions, leading to unhealthy dependence or excessively high moral expectations [1].
- In my opinion, the preliminary results show that "models without explicit empathy mechanisms will fail on these metrics", but do not show that "models with explicit empathy mechanisms can success".
- The response to alternative view 1 is not sufficient. The response is saying that "current alignments have not yet introduced empathy-related rewards". The lacking part is "if adding empathy-related rewards in RLHF or even adding empathy-related data in pre-training, whether LLMs can perform well on the proposed four metrics".
- The preliminary results risk potention circular reasoning. The paper advocates that models such as GPT lack empathy, but uses GPT-5.2 as a judge to evaluate the proposed metrics. The validity is limited. Large scale human evaluation is needed.
- The paper needs further discussion about alignment tax.

[1] Being Kind Isn't Always Being Safe: Diagnosing Affective Hallucination in LLMs. Sewon Kim, Jiwon Kim, Seungwoo Shin, Hyejin Chung, Daeun Moon, Yejin Kwon, Hyunsoo Yoon. EACL 2026 Findings.

**Support:**

2

---

> ### Author Rebuttal · Authors · 2026-03-31
>
> Thank you for recognizing the broad interest and timeliness of our position, as well as its potential benefits for context-sensitivity.
>
> > W1: Overemphasizing emotional resonance may lead to sensationalization or subjective bias in the news domain.
>
> We agree that this is critical in the news domain. Our goal is not to amplify emotion but to **calibrate affective salience**. As noted in Section 5.1, sensationalism is itself miscalibration.
>
> Conventional metrics can favor factually correct outputs that *mute accountability, urgency, or human impact* present in the source. Section 4 shows that strong semantic fidelity can still coexist with systematic attenuation of socially relevant affect.
>
> > W2: Risk of anthropomorphism and user over-reliance on perceived machine emotions [1].
>
> We agree that anthropomorphism is an important risk. Our goal is not to imply that LLMs possess real emotions, but to treat empathy as an **observable property of outputs**, akin to helpfulness or safety.
>
> As discussed in AV4, we avoid anthropomorphic assumptions by focusing on whether models preserve intention, affect, and context, and by decomposing failures into measurable mechanisms (SA, EGM, CA, LD). In this sense, we analyze **behavioral fidelity**, not emotional personhood.
>
> > W3: No positive evidence that models with explicit empathy mechanisms can succeed on the proposed metrics.
>
> Our experiments are designed to establish that **current models without explicit empathy mechanisms exhibit systematic failures**, motivating empathy as a first-class objective.
>
> While demonstrating improvements is beyond the scope of this position paper, emerging work provides supporting evidence that explicit mechanisms can help. For example, RLVER **[R1]** introduces verifiable emotion rewards and substantially improves empathy benchmarks, and PERM **[R2]** reports consistent gains and strong human preference.
> We will clarify this scope and better position our contribution.
>
> [R1] Wang et al. RLVER: Reinforcement learning with verifiable emotion rewards for empathetic agents. ICLR 2026.
>
> [R2] Wang et al. PERM: Psychology-grounded Empathetic Reward Modeling for Large Language Models. arXiv 2026.
>
> > W4: AV1's response is insufficient, as it does not discuss whether empathy-related rewards in RLHF or pre-training data would improve performance on the four metrics.
>
> We agree that directly testing whether empathy-oriented rewards improve all four mechanisms is important, but beyond this paper's scope.
>
> Prior work supports feasibility: RLVER **[R1]** shows gains from emotion-based rewards, and PERM **[R2]** improves both benchmarks and human preference.
> While not defined in SA/EGM/CA/LD, these results provide evidence that *empathy-aware objectives can yield measurable improvements*.
> We will clarify this limitation and position our framework as enabling such future evaluation.
>
> > W5: Circular reasoning-paper claims GPT lacks empathy but uses GPT-5.2 as a judge; Large-scale human evaluation is needed.
>
> Our metrics are not based on free-form judgments: each uses **explicit scoring rubrics**, and the judge LLM evaluates against predefined criteria rather than subjective preference. To address potential bias, we repeat evaluations with **Claude-Sonnet-4.5** and **Gemini-3-Pro** (**Table T1, see JCeE**). Inter-judge agreement is consistently high (Spearman: SA 0.835-0.948, CA 0.748-0.924), indicating stable relative rankings.
>
> We agree that human evaluation is important and will include this as future work to further validate the findings.
>
> > W6: Insufficient discussion of alignment tax.
>
> We agree that empathy as an explicit objective may incur alignment tax if applied naively. However, in human-centered settings, this trade-off is important: models that flatten urgency or distort user perspective are not fully aligned.
> We will clarify how multi-objective and mechanism-level alignment can manage this trade-off, making empathic fidelity controllable rather than costly.
>
> > Q1: Justify the definitions of the four metrics.
>
> Thank you for this question. SA measures **directional attenuation** relative to a reference affective intensity, so $(a-b)/a$ reflects proportional loss. In contrast, EGM captures **bidirectional mismatch**, so $(a-b)/(a+b)$ ensures symmetry and avoids privileging either text.
> In short, SA models reference-based attenuation, while EGM models symmetric mismatch, motivating their different normalizations. We will clarify this and summarize ranges in **Table T2**.
>
> **Table T2**: *Metric definitions, ranges, and interpretations.*
>
> |Metric|Range|Interpretation|
> |-|-|-|
> |SA|$(-\infty,1]$|0: no attenuation; 1: strong affect flattening; negative: more intense than reference|
> |EGM|$[0,1]$|0: close alignment in detail density; 1: severe mismatch|
> |CA|$[0,1]$|0: no conflict avoidance; 1: all responses avoid conflict|
> |LD|$[0,+\infty)$|0: no distancing difference; larger: more abstract and depersonalized|

---

> > ### Author Rebuttal · Reviewer_1X8y · 2026-04-01
> >
> > While I appreciate the authors' detailed rebuttal, some concerns remain unresolved.
> >
> > First (W2), the rebuttal dismisses the risks of anthropomorphism by claiming they only measure "behavioral fidelity." However, it is precisely this realistic behavioral mimicry that triggers unhealthy user reliance, a risk the paper still fails to meaningfully address.
> >
> > Second, regarding W3, the authors admit they lack empirical proof that models can actually succeed on their proposed metrics (SA, EGM, etc.). Arguing that current models fail does not prove that their proposed explicit mechanisms are viable or that their metrics are optimizable.
> >
> > Finally, regarding W5, the high correlation between different LLM judges only proves they share the same alignment-induced "affective flattening," not that the metrics align with true human evaluation.
> >
> > Therefore, I maintain my score. The paper identifies an interesting problem but lacks the necessary proof-of-concept and rigorous human evaluation to support its strong positional claims.

---

> > > ### Author Response · Authors · 2026-04-02
> > >
> > > We thank the reviewer for the careful reading and constructive follow-up. We address each point below and will revise the paper to make these aspects clearer.
> > >
> > > > First (W2), the rebuttal dismisses the risks of anthropomorphism by claiming they only measure "behavioral fidelity." However, it is precisely this realistic behavioral mimicry that triggers unhealthy user reliance, a risk the paper still fails to meaningfully address.
> > >
> > > We thank the reviewer for this important clarification. We agree that **realistic behavioral mimicry can indeed increase the risk of user over-reliance**, even if no anthropomorphic claims are made. Our intent is not to dismiss this risk, but to separate *what is measured* from *how it is deployed*.
> > >
> > > Our framework evaluates **empathic fidelity as an observable property of outputs**, but this does not imply that higher fidelity should be maximized without constraints. In deployment, such behavior should be **explicitly governed**, e.g., through calibrated expression, uncertainty signaling, and safety-aware interaction policies that prevent users from attributing internal states or authority to the model.
> > >
> > > We will revise the paper to make this distinction explicit: **behavioral realism is a capability to be controlled, not an objective to be blindly optimized**. This further reinforces our position that empathy should be incorporated under **multi-objective, mechanism-level alignment**, where risks such as over-reliance are explicitly managed rather than implicitly ignored.
> > >
> > > ---
> > >
> > > > Regarding W3, the authors admit they lack empirical proof that models can actually succeed on their proposed metrics (SA, EGM, etc.). Arguing that current models fail does not prove that their proposed explicit mechanisms are viable or that their metrics are optimizable.
> > >
> > > We thank the reviewer for highlighting this important point. We agree that **demonstrating end-to-end success is ultimately necessary**, and we will clarify that this paper does not claim to provide a complete optimization solution.
> > >
> > > However, we respectfully disagree that the absence of such results undermines the validity of our proposal. The core contribution of this work is to **identify and operationalize previously unmeasured failure modes** that are consistently exhibited by current models. Establishing that (i) these failures are systematic and (ii) they are measurable with structured signals is a prerequisite for optimization, not a consequence of it.
> > >
> > > Importantly, there is **emerging empirical evidence** that these signals are actionable: recent works (e.g., RLVER, PERM) show that introducing **emotion- or psychology-grounded reward signals** leads to consistent improvements in empathy-related benchmarks and human preference. While these works do not directly optimize SA/EGM/CA/LD, they provide **independent support that empathy-aligned objectives are learnable and optimizable**.
> > >
> > > We will revise the paper to make this positioning explicit: our framework provides a **diagnostic and objective foundation** that enables principled optimization, rather than claiming to have fully solved it.
> > >
> > > ---
> > >
> > > > Regarding W5, the high correlation between different LLM judges only proves they share the same alignment-induced "affective flattening," not that the metrics align with true human evaluation.
> > >
> > > We thank the reviewer for this important observation. We agree that **high inter-judge agreement alone does not guarantee alignment with human judgment**, and we will make this limitation more explicit.
> > >
> > > However, we respectfully clarify that the role of multi-LLM evaluation is to assess **measurement stability**, not to claim human-level validity. The consistently high correlations across diverse judges (GPT-5.2, Claude, Gemini) indicate that the metrics capture **systematic and model-agnostic patterns**, rather than artifacts of any single evaluator. This is a necessary first step to ensure that the signals are well-defined and reproducible.
> > >
> > > Importantly, our metrics are **not free-form preference judgments**, but are grounded in **explicit, mechanism-level criteria** (e.g., attenuation, mismatch, avoidance), which reduces but does not eliminate shared bias. This distinguishes our setting from purely subjective evaluation.
> > >
> > > We fully agree that **human validation is essential** to establish external validity. We will revise the paper to (i) position current results as **diagnostic proxies**, and (ii) explicitly outline human evaluation (e.g., expert annotation of affect preservation and user-perceived distortion) as a critical next step.

---

### Official Review · Reviewer_JCeE · 2026-03-12

**Significance:** 3
**Argument Clarity:** 3
**Rating:** 4
**Confidence:** 4

**Questions:**

Please refer to the weakness section.

**Alternative Views Section:**

Yes

**Compliance With Llm Reviewing Policy A Conservative:**

Affirmed.

**Discussion Potential:**

3

**Final Justification:**

The paper is insightful and valuable for the community. The authors’ rebuttal meaningfully addresses several of my concerns, and I believe that, with the proposed revisions fully incorporated, the paper would likely be suitable for acceptance. However, as these changes are not yet clearly reflected in the current rebuttal, I will maintain my current score.

**Paper Summary:**

This position paper argues that LLMs should incorporate explicit mechanisms for human empathy. They formalize empathy as an observable behavioral property: the capacity to model and respond to human perspectives while preserving intention, affect, and context, and identify four recurring mechanisms of empathic failure in contemporary LLMs–sentiment attenuation, empathic granularity mismatch, conflict avoidance, and linguistic distancing. And they suggest a new research agenda that treats empathic fidelity as a first-class optimization objective, supported by targeted alignment signals and specialized benchmarks that isolate these critical failures in human-centered settings

**Position:**

Yes

**Position In Title:**

Yes

**Related Work:**

3

**Strengths And Weaknesses:**

Strengths:
1. I like their concept of LLM empathy, which provides a logical decomposition into Cognitive, Cultural, and Relational components.
2. The paper identifies and formalizes four specific, measurable failure modes, which helps make the discussion more operationalizable.

Weaknesses:
1. Relying on one frontier LLM to judge the "empathy" of another may carry its own inherent biases or "alignment flattening" that the paper argues against.
2. They are treating empathy as a “first-class objective” alongside helpfulness and safety. However, it lacks a detailed discussion of potential trade-offs. For example, if a model preserves affective urgency, could this trigger safety filters or be perceived as sensationalist in contexts that require objectivity?
3. It is unclear whether the four proposed measurable mechanisms are sufficient to evaluate model empathy comprehensively. As the authors themselves acknowledge in AV4, empathy is a highly complex concept in human society, and reducing it to a small set of mechanisms may risk oversimplifying the phenomenon.

**Support:**

4

---

> ### Author Rebuttal · Authors · 2026-03-31
>
> Thank you for appreciating our decomposition of LLM empathy and our formalization of four measurable failure modes.
>
> > W1: Potential judge LLM bias and alignment flattening in evaluation.
>
> We mitigate this concern by guiding the judge LLM with detailed scoring rubrics based on predefined criteria (**Appendices B-D**), rather than free-form judgments.
>
> To further assess robustness, we repeat evaluations with **Claude-Sonnet-4.5** and **Gemini-3-Pro** (**Table T1**).
> Inter-judge agreement is **consistently high** (Spearman: SA 0.835-0.948, CA 0.748-0.924), indicating **stable** relative rankings.
> While absolute scores vary (e.g., Claude-Sonnet-4.5 and Gemini-3-Pro assign lower SA values than GPT-5.2), this reflects known scoring tendencies rather than framework instability.
> We will include these results and discuss plans for human evaluation to further validate the findings.
>
> **Table T1**: *Psychological counseling results for relational empathy on a constructed counseling-style dataset.*
>
> |Model|GPT-5.2||Claude-Sonnet-4.5||Gemini-3-Pro||
> |-|-|-|-|-|-|-|
> ||SA↓|CA↓|SA↓|CA↓|SA↓|CA↓|
> |GPT-5.2|0.529|0.833|0.281|0.900|0.420|0.833|
> |Claude-Sonnet-4.5|0.481|0.933|0.239|0.967|0.393|0.867|
> |Gemini-3|0.512|1.000|0.214|1.000|0.358|0.767|
> |Grok-4.1|0.515|1.000|0.352|1.000|0.403|1.000|
> |Llama-4|0.529|0.667|0.479|0.621|0.450|0.433|
> |DeepSeek-V3.2|0.530|0.700|0.338|0.700|0.403|0.333|
> |Qwen3-VL|0.431|1.000|0.141|1.000|0.314|1.000|
> |Kimi-K2|0.487|0.933|0.213|1.000|0.365|1.000|
> |GLM-4.7|0.509|0.967|0.222|0.967|0.368|0.967|
> |Doubao-Seed1.8|0.337|0.800|0.114|0.800|0.234|0.667|
>
> > W2: Insufficient discussion of trade-offs between empathy, safety, and objectivity.
>
> Thank you for this important concern. Our goal is not to maximize emotionality, but to **calibrate** affect, preserving contextually appropriate salience, abstraction, and relational stance.
> As noted in **Section 5.1** and **AV3**, empathy is intended to complement, not override, factuality and objectivity.
>
> Crucially, preserving affective urgency does not imply sensationalism; it avoids over-neutralization that can obscure accountability or human impact while remaining factually grounded.
> In safety- or objectivity-critical settings, empathy-aware alignment should operate under explicit constraints rather than as an unconstrained objective.
>
> We therefore advocate **multi-objective alignment with mechanism-level control**, making trade-offs among helpfulness, safety, and empathic fidelity explicit and tunable rather than implicit.
>
> > W3: Sufficiency of four mechanisms; risk of oversimplifying empathy.
>
> Thank you for raising this crucial point. We do not claim that the four mechanisms exhaust the full concept of empathy.
> Rather, we focus on a **tractable subset** of recurring, measurable, and practically consequential failure modes that are under-captured by standard metrics.
> Our framing treats empathy as an **observable behavioral property**, avoiding anthropomorphism while enabling operational evaluation.
>
> The four mechanisms are motivated by (i) synthesis of patterns observed in prior psychology and NLP work, and (ii) empirical evidence in Section 4 showing they already expose systematic failures invisible to conventional metrics.
>
> We agree that a more comprehensive account may require additional dimensions (e.g., multi-turn stability or personalization). We therefore position these mechanisms as an *initial, extensible basis* for evaluation and alignment, rather than a closed taxonomy.

---

> > ### Author Rebuttal · Reviewer_JCeE · 2026-04-02
> >
> > * I appreciate the authors’ clarification that the four mechanisms are not intended to be exhaustive, but the paper currently presents them as a central operationalization of empathy. It would benefit from a clearer positioning of these mechanisms as an initial, extensible framework, along with a more explicit discussion of what aspects of empathy remain unmodeled.
> > * Thank you for clarifying that empathy should be treated as a calibrated objective rather than maximizing emotionality. However, the discussion remains conceptual. Given that the paper advocates treating empathy as a first-class objective, a more concrete analysis of trade-offs (e.g., with safety, objectivity, or factuality) would significantly strengthen the argument.
> >
> > Overall, I still think the paper is insightful and valuable for the community. For this reason, I maintain a positive evaluation of the work.

---

> > > ### Author Response · Authors · 2026-04-02
> > >
> > > We thank the reviewer for the thoughtful follow-up and for maintaining a positive evaluation of our work.
> > >
> > > > I appreciate the authors' clarification that the four mechanisms are not intended to be exhaustive, but the paper currently presents them as a central operationalization of empathy. It would benefit from a clearer positioning of these mechanisms as an initial, extensible framework, along with a more explicit discussion of what aspects of empathy remain unmodeled.
> > >
> > > We thank the reviewer for this constructive suggestion and fully agree that the current presentation may overemphasize the four mechanisms as a complete operationalization.
> > >
> > > Our intent is to position them as an **initial, tractable, and extensible framework**, focusing on a subset of **recurring, measurable, and practically consequential failure modes** that are largely invisible to existing metrics. We will revise the paper to make this scope explicit and avoid any implication of completeness.
> > >
> > > We will also expand the discussion to clearly delineate **important aspects of empathy that remain unmodeled**, including:
> > > - multi-turn interaction dynamics (e.g., longitudinal trust and adaptation),
> > > - personalization and user-specific context, and
> > > - outcome-level effects on user perception and behavior.
> > >
> > > These additions will clarify that our framework is intended as a **foundation for systematic measurement and analysis**, rather than a closed taxonomy, and to better situate it within a broader research agenda on empathy-aware alignment.
> > >
> > > ---
> > >
> > > > Thank you for clarifying that empathy should be treated as a calibrated objective rather than maximizing emotionality. However, the discussion remains conceptual. Given that the paper advocates treating empathy as a first-class objective, a more concrete analysis of trade-offs (e.g., with safety, objectivity, or factuality) would significantly strengthen the argument.
> > >
> > > We thank the reviewer for this valuable suggestion and agree that a more concrete discussion of trade-offs would strengthen the paper.
> > >
> > > Our key point is that **empathy should not be optimized in isolation**, but incorporated within a **multi-objective alignment framework** alongside safety, factuality, and objectivity. In practice, the trade-offs arise at the **mechanism level**: e.g., preserving affective urgency (SA) must be bounded to avoid sensationalism, while reducing conflict avoidance (CA) must be constrained to maintain safety and non-harmful responses.
> > >
> > > We will also emphasize that **miscalibration itself is a failure mode**: both over-suppression (flattening) and over-amplification (sensationalism) degrade alignment. This framing makes empathy a **calibrated, context-dependent objective**, whose interaction with safety and factuality can be explicitly managed rather than implicitly traded off.
> > >
> > > These clarifications will ground our position in a more concrete and operational perspective while remaining consistent with the scope of a position paper.

---

### Official Review · Reviewer_UPEp · 2026-03-13

**Significance:** 4
**Argument Clarity:** 2
**Rating:** 4
**Confidence:** 3

**Questions:**

- How were the judge LLMs for the empathy failure metrics validated?
- While touched upon briefly in the paper, how does the framework for relational empathy relate to prior work on sycophancy? What are important considerations when identifying the boundary between sycophancy and empathy?
- How is SA measured in Section 4.3 without a reference text?
- Do different kinds of empathy require different alignment protocols?

**Alternative Views Section:**

Yes

**Compliance With Llm Reviewing Policy A Conservative:**

Affirmed.

**Discussion Potential:**

4

**Final Justification:**

I have increased my score with the promised revisions and inclusion of the clarifications and responses to my questions in the draft. In particular, the draft should address the issues with citations and justify the metric definitions and normalizations. Overall, the paper provides an interesting framework for empathy and will lead to valuable discussions.

**Paper Summary:**

- The paper argues that LLM development should explicitly consider empathy, rather than expecting it to arise as a byproduct of alignment protocols.
- The paper characterizes LLM empathy as the observable capacity of models to preserve user intention, affect, and context when responding.
- The paper presents a holistic conceptual framework for LLM empathy failure modes: sentiment attenuation, empathic granularity mismatch, conflict avoidance, and linguistic distancing. It then maps these failure modes to real-world tasks where cognitive, cultural, and relational empathy are important.
- The paper presents empirical evidence that current LLMs do (factually) well at tasks like text summarization but are susceptible to the aforementioned empathy failure modes.

**Position:**

Yes

**Position In Title:**

Yes

**Related Work:**

2

**Strengths And Weaknesses:**

*Strengths:*
- The paper presents a holistic framework for LLM empathy failure modes that is grounded in psychology literature. The paper operationalizes concrete metrics for the failure modes.
- The paper clearly outlines three dimensions of LLM empathy (i.e., cognitive, cultural, relational) and does a good job motivating why these dimensions are important in real-world tasks like text summarization and conflict mediation.
- The paper provides empirical evidence for its arguments by evaluating 10 LLMs on news generation, literary translation, and psychological counseling for empathy failures.
- The paper will inspire important discussion about conceptualizing LLM empathy more holistically and concretely, and is relevant to recent work on sycophancy (e.g., [1]).

[1] Cheng, M., Yu, S., Lee, C., Khadpe, P., Ibrahim, L., & Jurafsky, D. (2025). ELEPHANT: Measuring and understanding social sycophancy in LLMs. arXiv preprint arXiv:2505.13995.

*Weaknesses:*
- The examples in Figure 1 are not immediately understandable. The caption should elaborate on why each example captures the respective empathy failure mode. Furthermore, lines 75-77 state that these failure modes “arise systematically from prevailing training and alignment practices and recur across application domains” without citing evidence.
- The metrics in Section 2.2 (Eqs. 1-4) are not well-justified. Each metric uses a different normalization without sufficient explanation of what the metric represents in simple terms. The range of each metric is also difficult to interpret, which makes it hard to understand the magnitude of the results in Section 4. Additionally, some metrics depend on a reference text, which may not exist for in-the-wild LLM conversations.
- There are some claims in the paper that are justified with references that are not directly relevant. For example, in lines 120-121, the claim “this miscalibration arises because pretraining emphasizes language modeling and commonsense acquisition” cites [1, 2]. Lines 357-359 state “a key reason empathic failures persist is that current alignment pipelines treat empathy as an incidental style rather than a first-class optimization target,” citing [3]. Furthermore, Section 5.2 contains large blocks of citations without discussion of what statements they are supporting.
- Minor comment: Figure 1 is not central to Section 5.2 and is disproportionately large. In contrast, Table 1 is more central and could be better highlighted.
- In Section 5.2, the discussion of which failure modes are most pertinent to each dimension of empathy (e.g., “cultural empathy failures primarily arise through empathic granularity mismatch and linguistic distancing”) could benefit from further justification of why these failure modes are relevant (e.g., through a hypothetical example) and the other failure modes are less relevant.
- Section 4 should contain more experimental details, e.g., that the empathy failure metrics are generated using a judge LLM. Section 4 could also benefit from some qualitative examples where traditional performance metrics are high but there is a severe empathy failure.

[1] Wei, J., Bosma, M., Zhao, V. Y., Guu, K., Yu, A. W., Lester, B., ... & Le, Q. V. (2021). Finetuned language models are zero-shot learners. arXiv preprint arXiv:2109.01652.

[2] Ouyang, L., Wu, J., Jiang, X., Almeida, D., Wainwright, C., Mishkin, P., ... & Lowe, R. (2022). Training language models to follow instructions with human feedback. Advances in neural information processing systems, 35, 27730-27744.

[3] Rafailov, R., Sharma, A., Mitchell, E., Manning, C. D., Ermon, S., & Finn, C. (2023). Direct preference optimization: Your language model is secretly a reward model. Advances in neural information processing systems, 36, 53728-53741.

**Support:**

3

---

> ### Author Rebuttal · Authors · 2026-03-31
>
> Thank you for recognizing our structured view of LLM empathy, its three-dimensional framing, and our empirical analysis across tasks.
>
> > W1: Figure 1 needs a clearer explanation, and Lines 75-77 lack supporting evidence.
>
> We agree that Figure 1 can better convey the link between examples and failure modes and will revise the caption to make this mapping explicit.
>
> For lines 75-77, we will strengthen support with both prior work **[R1][R2]** and our results (**Tables 2-4**), which show the same four mechanisms recurring across news, translation, and counseling despite strong standard metrics.
>
> [R1] Kirk et al. Understanding the effects of RLHF on LLM generalisation and diversity. ICLR 2024.
>
> [R2] Xu et al. Consistency of Responses and Continuations Generated by Large Language Models on Social Media. arXiv 2025.
>
> > W2: Limited metric justification and reference-free applicability.
>
> We agree that the metrics need clearer justification and interpretation and will add intuitive explanations along with a summary of ranges and endpoint meanings (**Table T2, see 1X8y**). Regarding normalization, each metric is designed to reflect a specific distortion pattern (relative attenuation, mismatch, proportion, or structural shift), which motivates different normalizations. We will make this rationale explicit.
>
> For reference dependence, we agree that reliance on reference texts limits applicability in an open-ended setting. References are available for news generation and translation, while in counseling, we already use a **reference-free formulation** that treats the client narrative as the affective baseline. More generally, these metrics can also be extended to reference-free settings.
>
> > W3: Unclear relevance of citations.
>
> Thank you for pointing out this issue. [1] and [2] aim to describe pretraining objectives rather than direct evidence for empathic miscalibration, and will be replaced with more targeted evidence. Similarly, [3] will be clarified as illustrative of preference optimization and supplemented with more direct evidence on alignment underspecifying empathy.
>
> In Section 5.2, we will revise the presentation to more clearly connect each citation (e.g., ROUGE, chrF, COMET) to the specific claim that current benchmarks primarily emphasize semantic adequacy.
>
> > W4: Figure 1 is not central to Section 5.2; Table 1 could be better highlighted.
>
> Thank you for the suggestion. Figure 1 is intended to support Section 4 by summarizing the four mechanisms and their connections to the empirical tasks. We will clarify this role in the caption and make Table 1 more prominent.
>
> > W5: Limited justification for dimension-to-failure-mode mappings.
>
> We agree that the mappings in Table 1 need clearer justification. We will add brief explanations and illustrative examples to clarify why certain mechanisms are most diagnostic for each dimension. For example, replying to "I feel stuck and don't know why this hurts" with "Try making a to-do list" primarily reflects conflict avoidance, as the response bypasses unresolved emotional tension and shifts prematurely to advice. The core issue is not granularity mismatch, but the deflection of interpersonal uncertainty.
>
> > W6: Section 4 needs more experimental details and qualitative examples.
>
> Thank you for the suggestion. To improve clarity, we will add key experimental details from Appendices A-D to Section 4, including that metrics are computed with an LLM judge. To highlight the gap between conventional metrics and empathic fidelity, we will also include qualitative examples where standard metrics are high, but empathy fails (e.g., shifting from victim-specific suffering to generic advice or abstract discussion).
>
> > Q1: Judge LLM validation.
>
> We guide the judge LLM with detailed scoring rubrics based on predefined criteria (**Appendices B-D**), reducing subjective variation. For robustness validation, please refer to the reply of **JCeE W1**. We will include these results and discuss plans for human validation in future work.
>
> > Q2: Connection between relational empathy and sycophancy.
>
> We view relational empathy and sycophancy as **related but distinct**. Relational empathy aims to engage the user's perspective while keeping trust and appropriate interaction, whereas sycophancy involves unwarranted agreement that prioritizes user approval over truth or independent judgment.
>
> > Q3: SA measurement in Section 4.3 without a reference text.
>
> For counseling, no gold reference is available, so we use the **client narrative** as the SA reference (Appendix D), as it provides a clear affective intensity and emotional context for comparison.
>
> > Q4: Do different empathy dimensions need different alignments?
>
> Yes. Different kinds of empathy require **distinct** alignment protocols: cognitive empathy benefits from salience-aware signals, cultural empathy from culturally grounded preference data, and relational empathy from multi-turn supervision with feedback on interaction outcomes.

---

> > ### Author Rebuttal · Reviewer_UPEp · 2026-04-03
> >
> > Thanks for your thoughtful and detailed response! I would be happy to increase my score with the promised revisions and inclusion of the clarifications and responses to my questions in the next draft.
> >
> > W1. R1, R2, and Tables 2-4 seem to support that RLHF reduces output diversity and empathy-related issues appear across domains. However, they do not support the assertion that empathy-related issues arise systematically as the result of standard pretraining and alignment practices.
> >
> > W3. Could you please elaborate on "will be replaced with more targeted evidence?" Do you mean that you will use different references to support the claims in lines 120-121?

---

> > > ### Author Response · Authors · 2026-04-03
> > >
> > > Thank you very much for your encouraging feedback and for considering a score increase.
> > >
> > > We truly appreciate your thoughtful questions and constructive suggestions throughout the review process. They have been invaluable in helping us refine the claims, strengthen the evidence, and improve the overall clarity and positioning of the paper. We will carefully incorporate all promised revisions and clarifications in the next version.
> > >
> > > We are happy to further clarify the two points below.
> > >
> > > ---
> > >
> > > > W1. R1, R2, and Tables 2-4 seem to support that RLHF reduces output diversity and empathy-related issues appear across domains. However, they do not support the assertion that empathy-related issues arise systematically as the result of standard pretraining and alignment practices.
> > >
> > > Thank you for this careful observation. We agree that the current evidence ([R1], [R2], and Tables 2-4) primarily provides **indirect evidence** (e.g., reduced diversity under RLHF and cross-domain regularities), but does not by itself establish a strong causal claim that these issues arise systematically from standard pretraining and alignment practices.
> > >
> > > In the revision, we will **refine this statement to better match the evidence**. Specifically, we will (i) soften the causal phrasing (e.g., replacing "arise systematically'' with "consistently emerge under current training and alignment regimes''), and (ii) clearly distinguish between **empirical observations** (our experiments and prior findings on RLHF-induced homogenization) and **hypothesized mechanisms**.
> > >
> > > Our intent is not to overclaim causality, but to highlight a **robust and recurring empirical pattern** that motivates treating empathic fidelity as an explicit objective. We appreciate this suggestion and will revise the wording and evidence accordingly to make this distinction precise.
> > >
> > > ---
> > >
> > > > W3. Could you please elaborate on "will be replaced with more targeted evidence?" Do you mean that you will use different references to support the claims in lines 120-121?
> > >
> > > Yes, by "more targeted evidence," we mean that we will **replace and refine the current references** in lines 120–121 with citations that can directly support the specific claim being made, rather than using general background references.
> > >
> > > Concretely, we will include work such as "Does Pre-training Induce Systematic Inference? How Masked Language Models Acquire Commonsense Knowledge" (NAACL 2022), which shows that pretraining primarily captures **statistical co-occurrence and language patterns**, rather than deeper reasoning or calibrated understanding. This supports our argument that pretrained models may encode surface-level regularities without reliably preserving context-sensitive signals (e.g., affective salience or perspective), contributing to downstream miscalibration.
> > >
> > > We will also revise the surrounding text to clarify that pretraining is **not presented as a direct cause** of empathic failure, but as a **structural bias toward pattern completion and distributional smoothing**, which can interact with alignment objectives (e.g., preference optimization toward safe and neutral outputs) to produce the observed distortions.
> > >
> > > Overall, the revision will (i) use **more directly aligned references**, and (ii) present the claim as a **mechanistic hypothesis grounded in both prior findings and our empirical observations**, rather than an overstated causal assertion.

---

### Decision · Program_Chairs · 2026-04-30

**Decision:**

Accept (regular)

**Comment:**

After rebuttal the paper remains borderline, with ratings 3/4/4/5, and in particular Reviewer 1X8y remaining unconvinced by the lack of clear evidence that it's possible to improve on the empathy metrics defined in the paper, as well as the lack of human evaluation to validate LLM judge evaluations. Although those are valid concerns, I consider that (a) demonstrating improvements is beyond the scope of such a position paper, and (b) the high inter-judge agreement reported by the authors -- on well defined evaluation criteria -- is sufficient to build confidence that those results are based on real signal and not just some kind of judge bias. As a result, given that this paper exposes a novel blind spot of existing LLM evaluations, which is convincingly demonstrated empirically with well define evaluation metrics, I believe it is worth sharing with the ICML community.